# A Novel Terrestrial Rabies Virus Lineage Occurring in South America: Origin, Diversification, and Evidence of Contact between Wild and Domestic Cycles

**DOI:** 10.3390/v13122484

**Published:** 2021-12-11

**Authors:** Diego A. Caraballo, Cristina Lema, Laura Novaro, Federico Gury-Dohmen, Susana Russo, Fernando J. Beltrán, Gustavo Palacios, Daniel M. Cisterna

**Affiliations:** 1Instituto de Ecología, Genética y Evolución de Buenos Aires (IEGEBA), CONICET-Universidad de Buenos Aires, Ciudad Universitaria-Pabellón II, Buenos Aires C1428EHA, Argentina; 2Facultad de Ciencias Exactas y Naturales, Universidad de Buenos Aires, Buenos Aires C1053ABH, Argentina; 3Servicio de Neurovirosis, Administración Nacional de Laboratorios e Institutos de Salud (ANLIS), Instituto Nacional de Enfermedades Infecciosas, “Dr. Carlos G. Malbrán”, Av. Vélez Sarsfield 563, Buenos Aires C1282AFF, Argentina; clema@anlis.gob.ar (C.L.); dcisterna@anlis.gob.ar (D.M.C.); 4DILAB, SENASA, Av. Paseo Colón 367, Buenos Aires C1063ACD, Argentina; lnovaro@senasa.gov.ar (L.N.); srusso@senasa.gov.ar (S.R.); 5Instituto de Zoonosis “Dr. Luis Pasteur”, Av. Díaz Vélez 4821, Buenos Aires C1405DCD, Argentina; guryfe@hotmail.com (F.G.-D.); ferbelt@hotmail.com (F.J.B.); 6Department of Microbiology, Icahn School of Medicine at Mount Sinai, New York, NY 10029, USA; gustavo.palacios@mssm.edu

**Keywords:** rabies, nucleoprotein, phylogeny, dog-related, host shift, selection, recombination

## Abstract

The rabies virus (RABV) is characterized by a history dominated by host shifts within and among bats and carnivores. One of the main outcomes of long-term RABV maintenance in dogs was the establishment of variants in a wide variety of mesocarnivores. In this study, we present the most comprehensive phylogenetic and phylogeographic analysis, contributing to a better understanding of the origins, diversification, and the role of different host species in the evolution and diffusion of a dog-related variant endemic of South America. A total of 237 complete Nucleoprotein gene sequences were studied, corresponding to wild and domestic species, performing selection analyses, ancestral states reconstructions, and recombination analyses. This variant originated in Brazil and disseminated through Argentina and Paraguay, where a previously unknown lineage was found. A single host shift was identified in the phylogeny, from dog to the crab-eating fox (*Cerdocyon thous*) in the Northeast of Brazil. Although this process occurred in a background of purifying selection, there is evidence of adaptive evolution -or selection of sub-consensus sequences- in internal branches after the host shift. The interaction of domestic and wild cycles persisted after host switching, as revealed by spillover and putative recombination events.

## 1. Introduction

*Lyssavirus* (family *Rhabdoviridae*), which are etiological agents of a fatal infection of the nervous system of mammals, represent an excellent case study to gain insight into virus genetic diversity and disease emergence. Although several mammalian species are susceptible to infection, most lyssaviruses are only able to sustain onward transmission in a relatively low number of host species, indicating the presence of strong barriers to host shifting [1,2,3,4]. One exception to this general rule is the rabies virus (RABV), which is present worldwide and circulates in quite a diverse group of mammals including *Chiroptera* and *Carnivora*. Phylogenetically, the RABV is divided into two reciprocally monophyletic groups: a bat-related group confined to the American continent that circulates predominantly in chiropterans, with some circulation in skunks and raccoons in North America [5], and a dog-related lineage circulating globally among dogs and in wild carnivores restricted to specific geographical areas [6]. The circulation in wild carnivores is a clear example of human-mediated emergence, which now maintain RABV cycles posing a threat to these species, but also to dogs and humans.

In South America, at least three distinct dog-related RABV phylogenetic lineages have been reported. When typed with a panel of monoclonal antibodies directed against the viral nucleoprotein [7], they distribute among two antigenic patterns: Antigenic Variant 1 (AgV1), which circulates in dogs and foxes (*Urocyon cinereoargenteus*) in two disjunct areas: in Colombia and Venezuela [8], and in Perú, Bolivia, east of Brazil, Paraguay and north of Argentina [7,9,10]. The other is Antigenic Variant 2 (AgV2), which is distributed in Brazil, Paraguay, and Argentina [7,9,11,12]. AgV2 circulates predominantly in dogs and crab-eating foxes (*Cerdocyon thous*), but also among other mesocarnivores such as the hoary fox (*Lycalopex vetulus*) and the South American Coatí (*Nasua nasua*) [12,13,14,15,16]. Thus, AgV2 is an ideal model to study cross-species transmissions (CSTs) and host shifts given its greater promiscuity in terms of the number of stable or eventual hosts.

Although several studies have been published focusing on molecular epidemiology [11,13,15,17,18], dispersal patterns [19,20], and selection [12], these have been restricted to Brazilian lineages, and none of them included the complete diversity (even within Brazil) and geographic distribution. Herein, we present a study of the RABV AgV2 complete distribution, utilizing a data set of 237 full nucleoprotein sequences from Argentina, Brazil, and Paraguay detected during the period 1985–2017, with the aim to analyse its origins and diversification in space and time, inferring evolutionary patterns associated with host-switching, and gaining insight into the ecological interaction between wild and domestic cycles.

## 2. Materials and Methods

### 2.1. Sample Acquisition and Sequencing

A total of 55 dog-related RABV positive samples from Argentina, Brazil, and Paraguay isolated between the years 1989 and 2017 were typed by indirect immunofluorescence as described in Cisterna et al. [7] (Table 1). Briefly, acetone-fixed touch impressions of brain material were typed using a panel of eight monoclonal antibodies directed against the viral nucleoprotein (N-Mabs). Antigenic variant determination was performed following Díaz et al. [21].

We focused on the analysis of complete N ORF sequences, since this gene is the most thoroughly studied locus [11,13,14,15,16,17,19,20,22], and it is the only region of the genome that offers representatives of the entire geographic range and time span of the AgV2 lineage. Sequences obtained in this study were deposited in Genbank (Accession Numbers: OL451878-OL451932).

RABV N ORFs from 39 AgV2 and 16 AgV1 brain tissue samples were obtained by high-throughput sequencing (HTS). RNA was extracted with TriZol. RNA extracts were depleted of rRNA content and amplified by SISPA RT-PCR as described previously [23,24]. Amplification products were purified with AMPure XP magnetic beads (Agencourt) and used for Illumina library preparation with the Nextera XT DNA Library Prep Kit (Illumina) following the protocols provided by the manufacturer. Libraries were barcoded with non-overlapping dual indexes, pooled, and sequenced using the MiSeq Reagent kit v2 (Illumina) on an Illumina MiSeq instrument with a minimum of 2 × 250-bp reads.

The following pipeline was conducted to obtain full-genome consensus sequences. Open-source programs Cutadapt [25], Prinseq-lite [26], and Picard (http://broadinstitute.github.io/picard/ accessed on 20 March 2021) were used for adapter and SISPA primer removal, PCR duplicate removal, and quality filtering of the index (<30 Phred) and reads (<20 Phred). De novo assembly was performed with Ray [27], an algorithm that uses de Bruijn graphs to extend subsequences (seeds) into contigs using coverage information. Bowtie 2 [28] was used for aligning sequencing reads to the assembled contigs. Contigs were taxonomically checked in GenBank by BLASTn analysis (megablast and dc-megablast; e-value threshold: 1 × 10${}^{-04}$). Complete RABV ORFs were obtained following Ladner et al., 2015 [29]. Briefly, after adapter removal and quality filtering steps, reads were aligned to the genome scaffolds using Bowtie 2, after which duplicate removal was performed with Picard (http://broadinstitute.github.io/picard/ accessed on 20 March 2021), and finally, Samtools v0.1.18 [30] combined with custom scripts were used to generate a new consensus. Low-depth positions were excluded from the analysis; bases with Phred quality score ≥20 were kept, and a minimum of 3× read-depth coverage, were required for consensus calling; positions that did not meet such depth requirements were treated as “N”.

### 2.2. Bayesian Phylogenetic Analysis of Dog-Related Variants

A total of 118 Genbank dog-related RABV sequences were included representing all lineages and sublineages following Troupin et al. [6], with an enriched representation of South American samples (Appendix A). In particular, we selected representatives of AgV1 circulating in Brazil and Perú, and AgV2 from Brazil maximizing geographic and temporal coverage (Appendix A). Vaccine strains CVS and PV were also included. Among Brazilian AgV2 lineages, host-specific variants have been reported [11,12], and although spillovers have been reported in humans, cattle, and between the wild and domestic cycles, we opted to include sequences from the actual reservoir (i.e., dog in the domestic lineages, *C. thous*, and *L. vetulus* in the wild lineages). Eight sequences were annotated as isolated from “*Dusicyon* sp.” in Genbank. Taking into consideration that *Dusicyon* is an extinct lineage and that *L. vetulus* was previously assigned to this genus, these sequences could belong to that species. However, as pointed out by Carnieli et al. [11], the samples were isolated from foxes in the state of Paraíba, where *L. vetulus* is not normally found [31] and could belong to *C. thous*. However, there are records of *L. vetulus* in the state of Ceará [32,33,34], which borders the state of Paraíba, so we decided to maintain the ascription to this last species.

Nucleotide alignments were performed with Clustal Omega [35] (Appendix A). Nucleotide substitution models were estimated separately for each codon position using MrModeltest v2 [36] under the Akaike Information Criterion (correcting by the number of taxa). Selected models were K80+G, K80+I, and K80+G, for the 1st, 2nd, and 3rd codon positions, respectively.

A Bayesian phylogenetic analysis was carried out with MrBayes v3.2.7 [37] for 1.73 × 10${}^{8}$ MCMC generations sampling every 5 × 10${}^{3}$ generations. The potential scale reduction factor (PSRF) and the average standard deviation of split frequencies (ASDSF) were used for convergence diagnostics. The “burnin” phase was set up in the generation which fulfilled standard deviations lower than 0.01 and PSRF values of 1.00–1.02 for all estimated parameters, corresponding to 8.62% of the total run. Trees were visualized with Figtree 1.4.4 [38].

### 2.3. Phylogeographic Analysis of AgV2

For the phylogeographic analysis, we included the 39 AgV2 N gene sequences obtained in this study and retrieved from Genbank all AgV2 Brazilian samples (198) with corroborable geographic and temporal information (Appendix A). With this matrix, we ran an analysis in MrBayes v3.2.7 (5.6 × 10${}^{7}$ MCMC generations, sample frequency = 5000, “burnin” = 16.35%) (not shown), using a southern AgV1 sequence as outgroup (ARdgAA21SA2000). This analysis allowed us to have a first insight into the phylogenetic structure of the complete AgV2 lineage, after which we ran a subsequent analysis of 237 exclusively AgV2 sequences (1 × 10${}^{8}$ MCMC generations, sample frequency = 10,000, “burnin” = 10.53%) (Appendix A). A total of seven geographic regions were defined for inferring the ancestral location of each clade and for the most recent common ancestor (MRCA) of the AgV2 lineage: (1) North of Brazil, (2) Northeast of Brazil, (3) Midwest of Brazil, (4) Southeast of Brazil, (5) Paraguay, (6) Northeast of Argentina, and (7) Northwest of Argentina. The selected regions have intrinsic characteristics such as their biomes, weather, topography, hydrography, and vegetation, and as such are natural delimitations of the area where AgV2 occurs. This rationale has been adopted by the National Institute of Geography and Statistics of Brazil (https://www.ibge.gov.br/ accessed on 30 November 2021), and whose regional division is adopted in this study. Similarly, the northeastern and northwestern regions of Argentina are characterized by contrasting biomes: the Yungas (Mountain forests) and Puna (high-elevation grasslands), and the Alto-Paraná Atlantic forests (tropical rainforest) and Chacoan forest and Espinal (shrubland forest).

### 2.4. Temporal Analysis

To determine the degree of the clock-like structure in the AgV2 data set, we employed root-to-tip linear regression as available in the TempEst program. The analysis of the Bayesian phylogenetic tree based on the 237 AgV2 N gene sequences yielded a negative correlation between nucleotide divergence and time (Root-to-tip analysis), suggesting that the data contains little temporal signal, therefore being unsuitable for inference using phylogenetic molecular clock models. The observed pattern could be caused by intrinsic factors such as continuous fluctuations of the evolutionary rate through time, or by data quality issues such as errors in sampling dates annotation or incomplete/over-sampling [39]. Evidence for local clock models was further investigated selecting specific phylogenetic lineages. The R [40] packages phytools [41] and ape [42] were used in RStudio [43] to select each of the main lineages and identify candidate outliers. These analyses revealed also negative correlations between genetic divergence and time in two of the three lineages, and in the cases (and sublineages) where a positive correlation was found, a high number of outliers were identified, reinforcing the impossibility of analyzing this dataset under molecular clock models.

### 2.5. Network Analysis

To make a rapid inspection of geographic structure and time of isolation, a Median-joining network was inferred using PopART v1.7 [44] with the complete V2 dataset (237 sequences, Appendix A). The geographic analysis was carried out with a resolution at the state level. This analysis was used to test if variants showed geographical structuring at the regional level and, consequently, if this division was appropriate for Ancestral state reconstruction.

### 2.6. Ancestral States Reconstruction

Ancestral state reconstruction was performed with MBASR [45], which uses two R dependency libraries (ape and phytools) and MrBayes 3.2.7 to perform continuous-time Markov modeling against a tree’s topology and branch lengths, allowing for uncertainty in internal node states. A total of 1 × 10${}^{6}$ MCMC generations, (sampling frequency = 1000) were run. The tree topology and branch lengths were those inferred in the phylogenetic analysis based on the 237 AgV2 nucleoprotein sequences (Appendix A). Ancestral state reconstructions were performed for hosts and geographic regions. Localities were grouped into the seven geographic regions mentioned above and confirmed by the network analysis.

### 2.7. Recombination Analysis

To identify candidate recombinants, the alignment including the 237 AgV2 nucleoprotein sequences, was tested for recombination using the Recombination Detection Program RDP version 4 (RDP4) [46]. Default settings for all algorithms were applied (RDP, GENECONV, BootScan, Maxchi, Chimaera, SiSscan, and 3Seq), incorporating BootScan and SiSscan “primary scans” (the default option is that these two methods are only used to examine sequences in which recombination signals are detectable by other “primary scanning” methods). After removing potential recombinants (17 sequences), we reran the analysis checking that no new potential recombinants were detected. Phylogenetic conflict between the trees obtained with recombinant and non-recombinant segments is considered the gold-standard method to demonstrate recombination. Recombination was corroborated by running 100 Maximum Likelihood and Neighbor-joining bootstrap replicates in RDP4 of the potential non-recombined and recombined segments for each recombination event. If bootstrap support on the tree indicates that the recombinant sequences groups with one parental sequence for one segment and another parental for another segment, this is evidence of a statistically-supported phylogenetic recombination signal.

To assess possible historical recombination, we also tested a dataset containing all AgV2 N gene sequences as well as South-American AgV1 N gene sequences from the northern and southern distributions (11 sequences retrieved from Genbank, plus 16 sequences obtained in this study).

### 2.8. Selection Analysis

To assess if natural selection affected the evolution of AgV2 lineages, and particularly if host shifts were accompanied by changes in selective pressure, we employed codon-based and lineage-based Bayesian and Maximum Likelihood approaches to estimate rates of non-synonymous (dN) to synonymous substitutions (dS). Two datasets were analysed in the Datamonkey webserver [47]: (1) The 237 AgV2 N gene nucleotide sequence alignment, with its corresponding phylogenetic tree (Appendix A); (2) The recombination-free dataset (220 sequences), obtained after removal of potential recombinants. For this purpose a phylogenetic tree including non-recombinant sequences was inferred following the same partition and model selection schemes, using MrBayes (5 × 10${}^{7}$ MCMC generations, burnin = 10%).

Selection was inferred using the following codon-based methods: Single sites under selection were identified using Single Likelihood Ancestral Counting (SLAC) [48], Fixed Effects Likelihood (FEL) [48], adaptive Branch-Site Random Effects Likelihood (aBSREL) [49], Mixed Effects Model of Evolution (MEME) [50], as well as Fast Unconstrained Bayesian AppRoximation (FUBAR) [51]. Significance thresholds for selection tests were *P* ≤ 0.10 for SLAC, FEL, and MEME, *P*≤ 0.05 for aBSREL, and posterior probability ≥0.90 for FUBAR.

The aBSREL analysis was conducted in two ways: (1) an exploratory analysis was run, where all branches were tested for positive selection; (2) A priori selecting a set of “foreground” branches to test for positive selection. In this case, we selected both branches leading to the Wild Brazilian lineages (Brazil Wild A+B) as well as the MRCA of the AM3b lineage, taking into account that in one of these two nodes the host shift from dogs to wild mesocarnivores took place (see Results). We ran a third aBSREL analysis testing the whole Brazil Wild A + B clade, as well as the internal branch leading to AM3b. For the FEL analysis, we used this last setting as test branches and the rest of the AgV2 tree as reference.

Selection was also tested using tree-based methods at the Datamonkey server: RELAX [52] and BUSTED [53]. RELAX was used to determine whether selective strength was relaxed or intensified in the Wild A + B clade, relative to the rest of the tree. BUSTED tests for evidence of Episodic Diversifying Selection (EDS) in at least one site and one branch of the phylogeny. We ran this analysis testing the whole Brazil Wild A+B clade, as well as the internal branch leading to AM3b.

### 2.9. Amino Acid Ancestral Reconstruction

The amino acid ancestral reconstructions obtained in the SLAC test were analysed to find convergent or parallel changes for different clusters or lineages of the AgV2 variant. An amino acid replacement was operationally considered internal if it involved three or more descendant sequences, indicating that these are fixed mutations. Alternatively, an amino acid replacement was considered terminal if it was conserved in two or fewer descendant sequences, which denotes a transient (unfixed) amino acid polymorphism.

## 3. Results

### 3.1. Phylogenetic Analysis and Origins of the AgV2 Lineage

The Bayesian phylogenetic analysis revealed the existence of two different RABV terrestrial lineages occurring in Argentina showing total correspondence with antigenic variant typing (Figure 1). The samples typed as AgV1 (this study) belong to the AM5 lineage [6], which includes sequences from Perú, Brazil, and Argentina, also typed as AgV1. The samples typed as AgV2 (this study) belong to a clade that includes exclusively AgV2 variants from Brazil, Paraguay, and Argentina. These results confirm the monophyly and unique origin for AgV2. However, the relationship between the AgV2 clade and the two AgV1 clades occurring in South America is unclear, precluding the assessment of which of both lineages originated AgV2 (Figure 1). The Southern AgV1 clade is subdivided into two lineages; one composed of domestic dog sequences, and a separate basally splitting long-branch lineage corresponding to a fox from Perú (Figure 1, Appendix A).

Within the AgV2 clade, a novel terrestrial rabies lineage occurring in the Northeast of Argentina and Paraguay was found (Figure 1). This lineage splits basally from the clade including the Brazilian lineages AM3a and AM3b (Figure 1, [6]), and represents both domestic (dog) and wild canids (*C. thous* and *N. nasua*) from Argentina and Paraguay. Following the current classification scheme ([6], RABV glue http://rabv-glue.cvr.gla.ac.uk/ accessed on 30 August 2021), this lineage should be named AM3c. In Paraguay, sequences belonging to lineages AM3a and AM3c were found, evidencing that both cycles circulate in that country.

### 3.2. Phylogeographic Analysis

The Bayesian phylogeny of the 237 AgV2 sequences is congruent with the distinction of the lineages AM3a, AM3b, and AM3c (Figure 2 and Appendix A). The lineage AM3a is composed exclusively of domestic animals (dogs and cats) (Brazil Dog A), except for three spillover events involving *C. thous*. At least four sublineages can be identified within AM3a. The lineage AM3b is subdivided into three sublineages. A basally splitting sublineage is exclusively composed of domestic samples (Brazil Dog B). Two sister clades composed of mainly wild canid samples, although there are some spillovers to dogs, complete the AM3b lineage. The Brazil Wild A sublineage is characterized by relatively shorter branches and concentrates all the spillovers reported to dogs. The main host for this lineage is *C. thous*, although there are several isolates from *L. vetulus* (and “*Dusicyon* sp.”, which is probably the same species, as stated above). The Brazil Wild B sublineage has the longest branches compared with all other clades in the phylogeny. The AM3c lineage has no clear host structure, although most samples belong to domestic dogs. All AM3a, AM3b, and AM3c lineages and sublineages have posterior probabilities >0.9 (Appendix A).

The median-joining network revealed the existence of seven clusters which were corroborated in the phylogenetic analysis (Appendix A). These clusters are indicative of both the spatial and temporal structure of the AgV2 variant. The northeastern Dog-Wild cluster from Argentina (years 2001–2016) is more related to two Paraguayan samples (1991–1992) and followed by a northwestern subcluster from Argentina (1995–1996), which is extinct since terrestrial rabies was eliminated in the province of Tucumán. The most diverse region is the Northeast of Brazil, where at least four clusters can be differentiated. Two of these clusters are mainly composed of dog samples (years 2002–2009, and 2003–2005, respectively). One of these clusters is related to two wild-related clusters, also from the Northeast region of Brazil (years 2002–2011, and 2001–2009, respectively), showing an overlap in space and time of at least four lineages. Two remaining differentiated clusters complete the observed variation: one comprising dog samples from the Midwest and Southeast regions of Brazil (1987–2001), and the latter encompassing samples from the North-Northeast regions of Brazil (2003–2012).

The lineage AM3a can be subdivided into three subclades which reveal high levels of geographic structure (Figure 3). A first clade, composed exclusively of Brazilian samples from the neighboring Midwest and Southeast regions (in addition to a sample from Paraguay, reflecting the proximity of these territories). A second clade, which reflects high levels of geographic structuring, encompassing samples from the Northeast and North of Brazil. The basal split of the North Brazilian samples supports the hypothesis of the origin of this clade in this region and a secondary spread to the Northeast region. Remarkably, the totality of the Northeast samples corresponds to dogs from Maranhao, a Brazilian state which borders the North region, confirming a pattern of continuous diffusion of terrestrial RABV (Figure 3 and Appendix A). The third clade of the AM3a lineage is composed exclusively of Brazilian Northeast samples, except for a dog from Minas Gerais (Southeast), again a state bordering the Northeast region, confirming the continuous distribution of each lineage (Figure 3 and Appendix A). The MRCA of lineage AM3a likely originated in the Northeast of Brazil, supported also by the basal split of two samples from this region (unnamed clade within AM3a, Figure 2). The lineage AM3b originated in the Northeast of Brazil and occurs exclusively in that region. The MRCA of AM3a, AM3b was probably from the Northeast of Brazil, in concordance with their respective MRCAs.

The AM3c lineage shows a clear geographic structure. The most basally splitting clade corresponds to the northwestern Argentinian samples. The Paraguayan samples are sister to the northeastern Argentinian samples, reflecting their geographic proximity (Figure 3). The MRCA of all AgV2 lineages probably existed in the Northeast of Brazil (49.47% posterior probability), spreading to Argentina and Paraguay, to the North region of Brazil, and the Midwest and Southeast regions of Brazil in three independent waves mediated by domestic dogs (Figure 3).

### 3.3. Cross-Species Transmission Analysis

The three main AgV2 lineages, AM3a, AM3b, and AM3c, as well as their sublineages, show a clear predominance of whether domestic or wild species, which supports the notion of high host specificity in terrestrial RABV. However, as mentioned above, several spillover events could be identified within the three main lineages. It is noteworthy that these spillovers occurred between cycles co-occurring in time and place. For example, the three dog→*C. thous* spillovers located in the AM3a lineage were from the Northeast region of Brazil in the years 2005–2009, which coincides with the time of sampling of dogs in the same clade (Figure 2 and Appendix A). In one of the wild cycles in Brazil (Brazil WIld A), a total of nine wild→dog spillover events were identified. In contrast, in the remaining clade (Brazil Wild B), no such events were found, suggesting that the first would be a cycle occurring in a contact zone between dogs and wild canids, while the latter is likely to be a more internal -or isolated- cycle. It is notable that in both cycles *C. thous*→*L. vetulus*(or “*Dusicyon* sp.”) spillovers are observed. In the northeast of Argentina, a total of seven dog→*C. thous* and three dog→*N. nasua* spillover events were recorded. As expected, the ancestral host reconstruction of all main AgV2 clades reflected the dominant host. The ancestral host of the AM3b lineage is ambiguous being the domestic dog (45.22%) and *C. thous* (41.23%) nearly equiprobable species. The host shift should have taken place in the AM3b MRCA or, most likely, in the subsequent branch which leads to the Brazil Wild (A+B) lineage (Figure 2). Except for these two lineages, the domestic dog is the most probable host of all main clades, including the MRCA of AgV2. Taken together, these results support the origin of the AgV2 variant in dogs from the Northeast region of Brazil and the occurrence of a single host shift in the branch leading to the common ancestor of the two wild cycles circulating in the same region.

### 3.4. Recombination Analysis

No recombinants were found between AgV1 and AgV2 N gene sequences. A total of eight putative recombination events were detected among AgV2 sequences, three of which have strong signals of recombination and were corroborated phylogenetically (Figure 4, Appendix A). Two of such events occurred between different sublineages of the same cycle. Event 2, is the product of sequence exchange between members of the wild cycle, involving sequences from Brazil Wild A and Brazil Wild B sublineages. Event 4, involves exclusively sequences of the lineage AM3a (Brazilian domestic cycle; Brazil Dog A). Notably, the best-supported event (statistically significant by five different methods) is event 1, which involves sequences of the Brazil Wild B and members of the domestic cycle (Brazil Dog A). Three sequences were identified as descendants from this recombination event: BR9336CthBA2004, BR7578CthBA2005, and BR7579CthBA2005 (Figure 4 and Appendix A). This event accounts for the extremely long branches observed in these sequences and involves a track of 264 bp (bases 518–782 of the N gene) (Figure 2 and Appendix A). Although no *C. thous*→dog spillover events were found in this clade, these results would evidence the interaction between the Brazil Wild B sublineage and the less related domestic cycle.

Although the recombination events depicted varying levels of support across methods, we adopted a conservative criterion, which consisted in removing all candidate recombinants, after which we rerun the last analysis confirming that no additional events could be detected. A total of 17 candidate recombinants were excluded for subsequent phylogeny-based analysis. Notably, the 15 putative recombinants detected in the wild lineage belong exclusively to the Brazil Wild B clade (Appendix A). The two putative recombinants found in the domestic cycle derive from a common recombination event and belong to the Northeast clade of the AM3a lineage (Appendix A).

### 3.5. Selection Analysis

Codon-based selection analyses confirmed, as expected, that the N gene evolved under a background of strong purifying selection (Table 2, Figure 4). Five codons out of 450 showed evidence of positive selection, two of these occurring in the Brazil Wild A lineage, which was set as the test clade in the FEL analysis. According to lineage-based methods, no evidence of episodic diversifying selection or selection relaxation was found in the Brazil Wild clade (Table 2, but see the following item and Discussion). In the whole gene analysis (BUSTED), there is evidence that at least one site on at least one test branch has experienced diversifying selection, but this can be interpreted as a weak signal since it was not corroborated through any of the lineage-based methods.

Two amino acid replacements of the five positively selected codons were mapped in internal branches, while the other three occurred in terminal branches. An Ala→Ser substitution occurred in codon 77 in a branch leading to one of the Brazil Wild A subclades (Figure 4). The other amino acidic replacement occurred in codon 419, where a Ser→Leu substitution took place. These two amino acid replacements could be fingerprints of adaptation to *C. thous* after the domestic→wild host shift event.

When analyzing the complete dataset of 237 sequences, three branches were detected as evolving under positive selection. It is remarkable that all the sequences were previously identified as confirmed recombinants, highlighting the importance of detecting recombination prior to conducting dN/dS tests (Appendix A).

### 3.6. Amino Acid Replacements

A total of 131 amino acid replacements were inferred by ancestral reconstruction, 15 of which were convergent while four underwent inversions (Appendix A). These mutations occurred in 87 of the 450 residues of the N gene. The majority were terminal substitutions (73.3%), although there was a proportion of these changes in internal branches (26.7%). Internal replacements involved 3–95 descendant sequences, with a median value of 8. Fifteen terminal replacements could be associated with spillovers. In two spillovers (dog→cat and dog→*C. thous*) two simultaneous replacements occurred in different residues. Interestingly, in a *C. thous*→dog spillover, three amino acidic replacements took place, which would reflect a high rate of evolution in a terminal branch (but could also be attributable to sub-consensus variability, as discussed below). Remarkably, none of the internal replacements occurred in the branches involving the host shift (Figure 4). However, a significant number of replacements took place in internal branches of the AM3b lineage. At least four amino acid changes took place in two sublineages of the Brazil Wild A clade, while five replacements occurred in the branch leading to the Brazil Wild B clade (Figure 4). The latter is the branch with the highest number of amino acid replacements in the whole phylogeny, followed by the branch leading to AM3a and the previously mentioned Brazil wild A sublineage, with three substitutions each.

## 4. Discussion

### 4.1. Origins and Diversification of AgV2

Dogs are responsible for >99% of human rabies virus cases worldwide and are the main vector of dog-related RABV [54]. One of the main outcomes of long-term RABV maintenance in dogs was the establishment of variants in a wide variety of mesocarnivores [55]. This is evidenced at the phylogenetic level, where it is verified that wild species lineages split from different internal clades of the Cosmopolitan group of dog-related RABV [6]. During this process, a RABV variant establishes a species-specific relationship with its new host, compartmentalizing the disease within geographically discrete enzootics, originating a novel variant that may interact with other susceptible mammal species at its proximities [56]. The establishment of a new cycle will depend on ecological and host species traits.

In this study, we present the most comprehensive RABV AgV2 phylogenetic and phylogeographic analysis confirming previous findings and adding new results contributing to a better understanding of the origins, diversification, and the role of different host species in the evolution and dissemination of this dog-related variant. Although the lineage of which AgV2 descends is not clear, we confirm that this variant is monophyletic. Even though it has been reported that Brazilian lineages were monophyletic [11,12], the inclusion of Paraguayan and Argentinian samples could alter this hypothesis. These samples pertain to a novel basally splitting lineage, AM3c. We confirmed the existence of two main lineages within Brazil (AM3a and AM3b), one of which is composed of exclusively domestic sublineages (Brazil Dog A) [11,15,16], while the other consists of a basally splitting domestic clade (Brazil Dog B) [15], and two sublineages circulating in *C. thous* (Brazil Wild A, Brazil Wild B), which overlap in space and time [11,12,16]. The sequences belonging to the Brazil Dog B lineage have been included only in their initial publication [15]. We point out the need of including the complete dataset of AgV2 sequences since these are the only representatives of the domestic cycle of the AM3b sublineage, which has been previously interpreted as a pure wild lineage [16].

The AgV2 variant originated in dogs in the northeast of Brazil and dispersed to the south and to the west as a result of urbanization and human migrations [11,12,15] (this study). In a previous study, the border of the Brazilian states of Paraíba and Pernambuco was identified as the most probable place of origin of Brazilian AgV2 [20]. This is congruent with the reconstruction of the node (AM3a, AM3b) obtained in the present study (Figure 3). After the initial dispersal, some lineages would have reversed the direction of the spread into contiguous regions, showing that this process was not unidirectional.

### 4.2. Cross-Species Transmission

To understand the different epidemiological scenarios of rabies in wild carnivores, a useful theoretical framework was provided based on the metapopulation concept [57], differentiating the processes of maintenance and persistence. Maintenance is the indefinite transmission of a virus through members of a local host population, while persistence implies the long-term and continuous presence of disease within a metapopulation. From a metapopulation point of view, successful rabies persistence would require the presence of rabies at any time at least in one local population, which is a more probable situation compared to maintenance. This perspective could be extended to the interspecific level: some species act as maintenance host populations; while others may maintain the disease locally, but after depletion and renewal of host local populations, may require contact with a species acting as maintenance host. These two different patterns can be tested from a molecular epidemiology approach: rabies virus from non-maintaining species are phylogenetically similar to the maintenance hosts sequences and do not fall into specific lineages, while established maintenance cycles can be evidenced as distinct species-specific monophyletic lineages. Under this framework, we can assert that both dogs and *C. thous* are actual reservoirs for RABV AgV2, the former being an urban and the latter a sylvatic reservoir. However, although *C. thous* is able to maintain successful transmission cycles in Brazil [11,12] (this study), it is probably a dead-end or is capable of maintaining short transmission cycles in Argentina, where domestic dogs are the only confirmed reservoirs. The same occurs with *L. vetulus* (and “*Dusicyon* sp.”), *N. nasua*, and domestic cats, which can be occasional hosts but not reservoirs, as denoted by the lack of host-specific monophyletic groups that would represent prolonged infection transmissions between members of the same species (Figure 2).

### 4.3. Host Shifting and Adaptation

The rabies virus is characterized by an evolutionary history dominated by host shifts within and among bats and carnivores [58,59,60]. In such events, minor or more extreme fitness variations are expected to emerge, determined by physiological and ecological differences between donor and recipient host species [61]. Adaptation to a new reservoir can occur after host shifting, but selection may also act over a repertoire of previously existing adaptations, as demonstrated by deep-sequencing of skunk associated variants implicated in an outbreak in foxes in North America [62], or the analysis of the determinants of bat RABV causing successful infection in carnivores [58]. Although amino acid replacements and episodic bursts of positive selection were detected in several RABV viral proteins associated with host shift events [1,2,58,59], these occurred through unique mutations, limiting the utility of past host shifts for predicting the evolutionary dynamics of future emergence. The single host shift that took place in the AgV2 variant originated the wild lineage circulating in *C. thous*, with subsequent spillovers to other wild canids. In agreement with the general pattern found in dog-related variants ([6] and references therein), the N gene evolved in a background of purifying selection (Figure 4). Our analysis showed that positive selection occurred after but not before host shifting. The two sites under positive selection in the wild lineage were mapped in internal branches of the Brazil Wild A clade (Figure 4). These could represent fingerprints of post-emergence adaptations, given that none of these amino acid changes experienced any further mutation, as documented in bats [59] and bats-to-carnivores CSTs [58]. Several amino acid replacements were mapped along branches leading to two Brazil Wild A sublineages, while the one leading to Brazil Wild B is the branch with the highest number of amino acid substitutions in the tree (Figure 4). These events may also represent post-host shift adaptations that have not been identified as positively selected sites through the analysis of dN/dS. False negatives were proposed in a previous paper where two parallel substitutions identified in independent host jumps from dogs to ferret-badgers, were undetectable through dN/dS tests [6]. The existence of two basally-splitting, independent wild lineages enables the possibility to test for parallel or convergent evolution after host shifting. Notably, there are no parallelisms nor convergences between the 16 internal amino acid substitutions that took place in the Brazil Wild A and Brazil Wild B lineages, evidencing the lack of a common mechanism driving the adaptive evolution of this RABV variant after host shifting, as reported in previous CST studies in skunks, foxes, and bats [58,59]. The majority of convergences occurred between terminal branches, or between one terminal and one internal branch (Appendix A). This observation, together with the presence of multiple terminal inversions supports the existence of sub-consensus sequences which may coexist in host populations [62]. This mechanism may also explain the presence of multiple substitutions occurring in terminal branches after spillovers (Appendix A), which are likely to be the result of the selection of pre-existing undetected variants instead of the consequence of extremely high rates of evolution in a single branch. Taken together, our results support the absence of pre-host shift adaptations but support the occurrence of non-convergent post-host shift adaptations which took place independently along the wild sublineages. However, we cannot rule out the existence of these variants at a sub-consensus level prior to the host shift.

Reinforcing the hypothesis of adaptation after host shifting, our analysis shows contrasting branch lengths between lineages. The total length of the tree is 0.0747 substitutions per site (this value is overestimated since the existence of recombinants yields artificially longer branches), which is congruent with a context of strong purifying selection (Figure 4). Both internal and terminal branch lengths are longer in the wild cycle, which is compatible with the occurrence of diversifying selection. However, there is no evidence of overall purifying selection relaxation in the wild lineage, and other factors could also explain the observed pattern. Ecological differences among host species are likely to influence observed branch lengths [11,12]. Whereas dogs can reach high population sizes and ranges in urban areas, *C. thous* is a territorial species that can live in packs but hunts alone [63]. Moreover, dogs disperse almost unrestrictedly since they are moved by humans (also feral dogs disperse), while the habitat of *C. thous* is determined by its ecological niche, and thus has lower dispersion rates [20]. Populations of *C. thous* are separated by longer distances, and connectivity is hampered by habitat fragmentation. Both factors contribute to higher expected local extinctions and lower levels of RABV homogenization. Vaccination may also influence the different patterns among wild and domestic clades, eliminating a large number of rabies lineages from the dog population [64].

### 4.4. Interaction between Domestic and Wild Cycles

Two lines of evidence account for ongoing interaction between the domestic and wild cycles: spillovers and the presence of potential recombinants. Several spillovers were corroborated between geographically overlapping lineages. Both dog→*C. thous* and *C. thous*→dog spillovers were found in the northeast of Brazil, which demonstrates the close contact between domestic and wild canids in this region (Figure 2). Notably, the Brazil Wild A clade concentrates all *C. thous*→dog spillovers (9 cases), while there are no such events in the Brazil Wild B lineage. In addition, a total of 10 domestic→wild spillovers were found in the Argentinian clade, denoting high levels of contact between dogs and wild mesocarnivores. Although there is no evidence of a wild cycle in this region, the existence of *C. thous* cycles in Brazil together with the high levels of dog-related infections leads to considering this species as a potential reservoir in Argentina.

Recombination is a common event in RNA viruses, and its detection is crucial to minimize the disruptive impact that this phenomenon can have on subsequent phylogeny-based analyses of molecular evolution [65,66]. Furthermore, recombination between viruses can reveal otherwise undetectable ecological links among non-related lineages [67,68]. In the rabies virus, recombination has been demonstrated to occur among Arctic-like viruses from Pakistan and India, cosmopolitan viruses from Africa, and Arctic group viruses [69], as well as among Chinese isolates [70,71]. These events were reported to occur in the N gene and its 5’ untranslated region, and the L gene, between closely related isolates (intraclade) and less related sequences (interclade). Many of these events were found in several isolates, evidencing the common ancestry of recombinants. Remarkably, the majority of recombination breakpoints reported occurred exclusively within RABV ORFs [60,69,71,72] or at least one of them involved an in-frame breakpoint [70]. This suggests that recombination, although to a lesser extent compared with mutation rate, can contribute to rapid amino acid replacement in RABV [70,72]. In addition, historical recombination events were reported, such as the exchange of G gene segments between carnivore and American bat RABV, which is present in raccoon RABV (RRV) and south-central skunk RABV (SCSKV) [60]. The occurrence of recombination between distantly related viruses suggests that genome compatibility is likely not the principal barrier to this phenomenon, and low apparent recombination is better explained by virus ecology rather than by molecular limitations or infection pathogenesis [69]. Homologous recombination (HR) has been historically thought to be rare or absent among negative-strand RNA viruses [73,74]. However, as mentioned above, there is robust evidence supporting the notion that, although infrequent, HR contributes to RABV diversity and evolution. Hence, recombination detection should be carried out mandatorily in RABV, not only for the correct assessment of selection but also because the presence of recombinants may lead to erroneous molecular typing in diagnostic laboratories. Molecular typing in RABV surveillance is generally based on short sequences targeting the nucleoprotein gene. If recombination takes place within the sequenced fragment, it could lead to incorrect RABV variant identification. As pointed by He and collaborators [70], given that recombination can lead to the emergence of novel pathogens with unpredicted epidemiological results, RABV recombinants should be taken into consideration in RABV surveillance. An example of these unexpected results is the polio epidemics caused by the recombination of a vaccine strain with endogenous enteroviruses [75].

In our analysis, five putative recombinants were found, originating from three independent recombination events (three sequences derive from a single recombination event) (Figure 4). In a previous study of recombination in RABV, three breakpoints were found in the N ORF originating mosaic sequences, located in positions 179, 597, and 890, overlapping with the three putative recombinant segments found in this study [70]. The three potential recombination events were corroborated by statistically incongruent phylogenetic trees based on major and minor parental sequence segments, which is considered the gold-standard method to demonstrate recombination. The detection of three putative recombination events among 237 sequences spanning the whole AgV2 diversity is in agreement with previous studies that state that recombination in RABV does occur, but is infrequent [72]. As mentioned, recombination is detected by the analysis of divergent sequence segments that produce incongruent phylogenies. But phylogenetic tree incongruence along a sequence alignment could be the product of convergent selective pressures at the molecular level. The quantification of synonymous and non-synonymous substitutions enables the distinction between recombination and convergent evolution [76]. If the putative recombining segment has predominantly non-synonymous substitutions which are shared with the minor parent lineage, it is likely to be the product of convergent evolution. Contrarily, if the putative recombining segment has a higher proportion of synonymous substitutions, it would be not attributable to any selective process, and hence, would be evidence of recombination. Of the three recombination events, we analysed event 1 and event 4, because both breakpoint positions were unambiguous. In event 1, in the recombinant segment (positions 518–782) the recombinants and the minor parental sequence shared one non-synonymous substitution and 19 synonymous substitutions. In event 4 (positions 840–1254), the ratio was 4:13 (non-synonymous:synonymous substitutions). The clear predominance of synonymous substitutions allows discarding the hypothesis of convergent evolution in these sequences.

The sequences involved in the recombination events were retrieved from Genbank and were obtained by a single laboratory, so we cannot completely rule out the possibility of contamination. However, it is noteworthy that in the recombined segments the potential recombinants share mutations with respect to the minor parental sequences. As proposed in the guidelines for recombination identification in negative-stranded viruses, evidence that recombinant sequences form a distinct circulating lineage (Figure 2) and have been transmitted among multiple individuals in a population, are strong indications of authentic recombination [74]. One of these events (event 2) occurred between members of Brazil Wild A and Brazil Wild B. A second putative recombinant (event 4) is the product of the interaction between Brazilian Northeast and North-Northeast domestic clades. These two first events took place between closely related lineages occurring in the same host species and the same geographic region. The most relevant putative recombination event reveals the interaction between wild and domestic cycles (event 1). Notably, the recombinants belong to Brazil Wild B, where no *C. thous*→dog spillovers were found, showing that in this lineage there is also some degree of contact with dogs (Figure 2 and Figure 4). However, given the absence of spillovers in this clade, it is likely that this lineage is more internalized, occurring in less anthropic habitats, and, thus, has reduced interaction with domestic dogs.

### 4.5. Conclusions

In this study, we have inferred the origins and diversification of the AgV2 RABV variant, showing that it originated in dogs in the northeast of Brazil, while in the same region, a host shift to *C. thous* took place. This variant was disseminated through Brazil, Argentina, and Paraguay, originating three lineages: AM3a, AM3b, and AM3c. All this process occurred in a background of purifying selection, although some sites may have gone through adaptive evolution -or selection of sub-consensus sequences- in internal branches after the host shift. The interaction of domestic and wild cycles persisted after host switching, as revealed by the presence of putative recombinants and by spillover events. As argued elsewhere [69,77], the potential for RABV for host-shifting (and for other processes such as recombination), would depend more on ecological factors (such as sympatry) than on phylogenetic closeness.

## Figures and Tables

**Figure 1 viruses-13-02484-f001:**
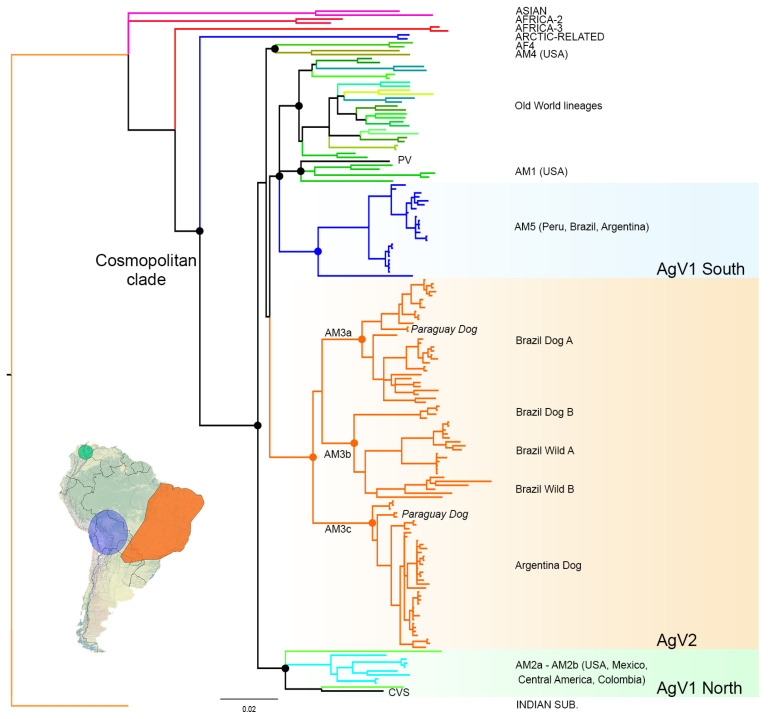
Bayesian phylogenetic tree of 174 dog-related RABV lineages. Circles indicate relevant internal nodes with posterior probability values >0.9. The scale bar represents substitutions per site.

**Figure 2 viruses-13-02484-f002:**
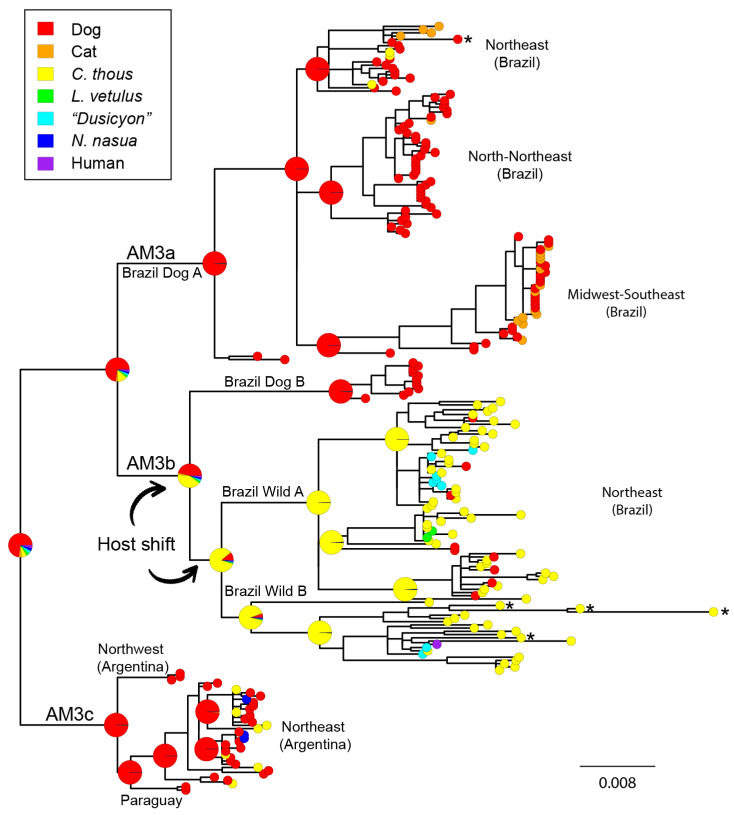
Bayesian ancestral host reconstruction of 237 AgV2 complete N gene sequences. The phylogenetic tree covers the complete distribution, host species, and temporal ranges of this variant. Colored tip nodes show the host species of each sample. Pie charts are shown for relevant nodes and represent uncertainty in ancestral states reconstruction. Upper-left: Reconstruction of the place of origin, initial spread, and secondary spread of AgV2. The scale bar represents substitutions per site. Asterisks denote recombinants for the three corroborated events.

**Figure 3 viruses-13-02484-f003:**
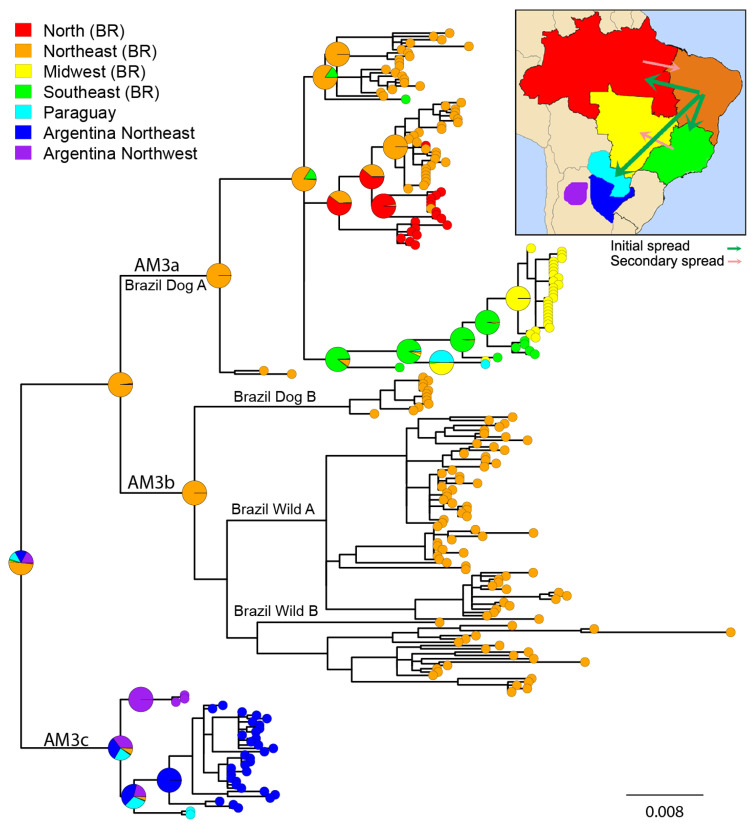
Bayesian ancestral geographic reconstruction of 237 AgV2 complete N gene sequences. The phylogenetic tree covers the complete distribution, host species, and temporal ranges of this variant. Colored tip nodes show the place of origin of each sample classified into four Brazilian regions, Paraguay, and northeast and northwest of Argentina. Pie charts are shown for relevant nodes and represent uncertainty in ancestral states reconstruction. Upper-right: Reconstruction of the place of origin, initial spread and secondary spread of AgV2. The scale bar represents substitutions per site.

**Figure 4 viruses-13-02484-f004:**
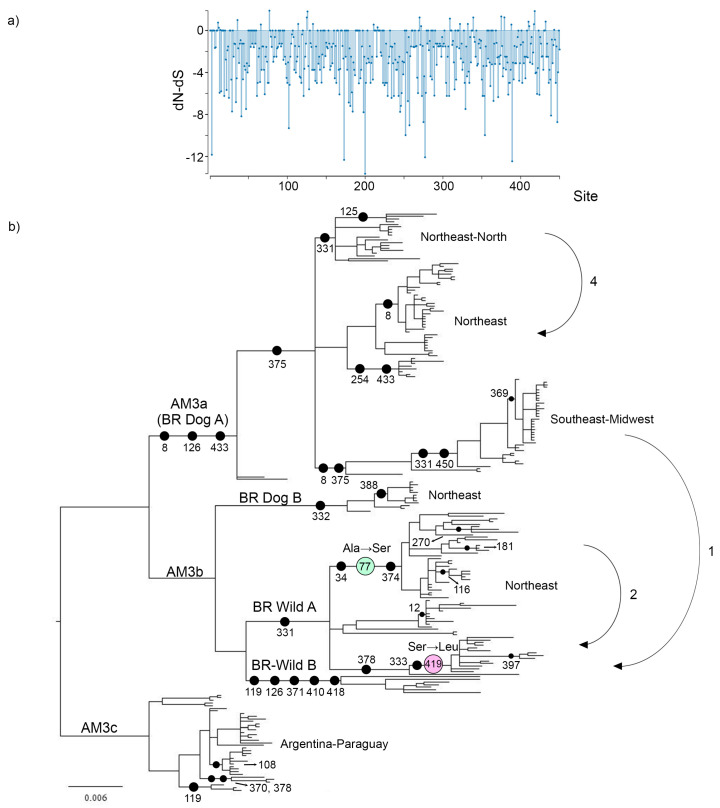
Selection and recombination events in the AgV2 lineage. (**a**) Estimated dN-dS ratios for the RABV N gene codons in the AgV2 lineage with the SLAC method. (**b**) Phylogenetic tree showing positively selected sites in the wild cycle and amino acid replacements along the entire phylogeny. Color-circled numbers represent positively selected codons identified by different methods (ancestral→derived states are shown). Black circles represent amino acid replacements. The numbers below these circles indicate the residue position. Curved arrows indicate lineages involved in the three bioinformatically confirmed recombination events (1, 2, and 4).

**Table 1 viruses-13-02484-t001:** Rabies virus nucleoprotein gene sequences obtained in this study.

No.	Species	Country	Province	Locality	Year	Variant	Genbank Accession Number
AA35	Dog	Argentina	Tucumán	Concepción	1995	AgV2	OL451878
AA38	Dog	Argentina	Tucumán	Aguilares	1996	AgV2	OL451879
AA39	Dog	Argentina	Tucumán	Chicligasta	1996	AgV2	OL451880
552	*C. thous*	Argentina	Chaco	El Espinillo	1997	AgV2	OL451910
AA53	Dog	Argentina	Formosa	Pozo del Tigre	2000	AgV2	OL451881
AA71	Dog	Argentina	Formosa	Pirané	2000	AgV2	OL451882
AA112	Dog	Argentina	Chaco	Ciervo Petizo	2001	AgV2	OL451883
AA120	Dog	Argentina	Chaco	Pampa Almirón	2001	AgV2	OL451884
A119	*C. thous*	Argentina	Chaco	Pampa Almirón	2001	AgV2	OL451912
A166	Dog	Argentina	Formosa	Potrero Norte	2002	AgV2	OL451909
AA148	Dog	Argentina	Chaco	Pirané	2002	AgV2	OL451913
AA474	*C. thous*	Argentina	Chaco	Presidencia Roca	2004	AgV2	OL451885
A696	*C. thous*	Argentina	Chaco	Resistencia	2006	AgV2	OL451911
AA774	Dog	Argentina	Formosa	Gran Guardia	2007	AgV2	OL451886
AA881	Dog	Argentina	Formosa	Palo Santo	2007	AgV2	OL451914
AA886	*C. thous*	Argentina	Formosa	Colonia Da Prato	2008	AgV2	OL451887
AA888	Dog	Argentina	Formosa	NA	2008	AgV2	OL451888
AA911	Dog	Argentina	Chaco	Pampa del Indio	2008	AgV2	OL451889
A874	Dog	Argentina	Chaco	Capitán Solari	2008	AgV2	OL451906
AA929	*N. nasua*	Argentina	Chaco	Laguna Limpia	2009	AgV2	OL451890
AA957	Dog	Argentina	Chaco	Colonia Elisa	2009	AgV2	OL451891
AA959	Dog	Argentina	Chaco	Colonia Elisa	2009	AgV2	OL451892
AA966	*N. nasua*	Argentina	Formosa	Palo Santo	2009	AgV2	OL451893
AB124	Dog	Argentina	Formosa	Comandante Fontana	2012	AgV2	OL451894
AB160	Dog	Argentina	Formosa	Ibarreta	2013	AgV2	OL451895
AB193	*C. thous*	Argentina	Chaco	La Verde	2013	AgV2	OL451896
AB353	Dog	Argentina	Formosa	Ibarreta	2013	AgV2	OL451897
AB356	*C. thous*	Argentina	Formosa	El Colorado	2013	AgV2	OL451898
AB357	Dog	Argentina	Formosa	Laishi	2013	AgV2	OL451899
AB361	*N. nasua*	Argentina	Formosa	Mariano Boedo	2013	AgV2	OL451900
AB362	Dog	Argentina	Formosa	Comandante Fontana	2013	AgV2	OL451901
AB118	Dog	Argentina	Formosa	S/D	2015	AgV2	OL451902
AB122	Dog	Argentina	Chaco	Puerto Tirol	2015	AgV2	OL451903
739	Dog	Argentina	Chaco	Pampa Almirón	2016	AgV2	OL451904
682	Dog	Argentina	Chaco	Laguna Blanca	2017	AgV2	OL451905
257	Dog	Brazil	Mato Grosso do Sul	Pantanais	1989	AgV2	OL451915
352	Dog	Paraguay	Itapúa	Encarnación	1991	AgV2	OL451908
353	Dog	Paraguay	Itapúa	Encarnación	1991	AgV2	OL451916
369	Dog	Paraguay	Itapúa	Encarnación	1992	AgV2	OL451907
AA42	Dog	Argentina	Salta	Orán	1999	AgV1	OL451918
AA47	Cat	Argentina	Salta	Orán	1999	AgV1	OL451919
AA50	Cat	Argentina	Salta	Orán	1999	AgV1	OL451920
AA21	Dog	Argentina	Salta	Hipólito Yrigoyen	2000	AgV1	OL451917
AA126	Dog	Argentina	Salta	NA	2002	AgV1	OL451921
AA178	Dog	Argentina	Salta	Aguaray	2002	AgV1	OL451922
AA180	Monkey	Argentina	Salta	Tartagal	2002	AgV1	OL451923
AA183	Dog	Argentina	Salta	Tartagal	2002	AgV1	OL451924
M863	Dog	Argentina	Jujuy	Palpalá	2003	AgV1	OL451929
M870	Dog	Argentina	Jujuy	El Carmen	2003	AgV1	OL451930
M1009	Dog	Argentina	Jujuy	S.S. de Jujuy	2003	AgV1	OL451932
M632	Dog	Argentina	Jujuy	S.S. de Jujuy	2005	AgV1	OL451926
M654	Dog	Argentina	Jujuy	S.S. de Jujuy	2005	AgV1	OL451928
M639	Dog	Argentina	Jujuy	S.S. de Jujuy	2006	AgV1	OL451927
M876	Dog	Argentina	Jujuy	S.S. de Jujuy	2006	AgV1	OL451931
M297	Dog	Argentina	Salta	Orán	2015	AgV1	OL451925

**Table 2 viruses-13-02484-t002:** Results of multiple analyses of selection in the N gene of the AgV2 recombination-free dataset (220 sequences). (**a**) Significant positively and negatively selected sites detected by site-based methods FEL, SLAC, FUBAR, and MEME. (**b**) Analysis of evidence of relaxation (Relax) or Episodic Diversifying Selection (aBSREL, BUSTED).

(a) Site-Based Methods					
**Length (Codons)**	**Type of Selection**	**FEL**	**SLAC**	**FUBAR**	**MEME**
450	Positive	2	0	0	3
	Negative	193	162	311	-
Codons		77,339	-	-	158, 419, 443
**(b) Branch-Based Methods**					
**Method**	**Test Group**		**Reference Group**	**Result**	
aBSREL	All AgV2		-	No EDS (*p* ≤ 0.05)	
aBSREL	AM3b MRCA + Brazilian Wild A + B MRCA		-	No EDS (*p* ≤ 0.05)	
aBSREL	AM3b MRCA + Brazilian Wild A + B		-	No EDS (*p* = 0.0057 ≤ 0.05)	
Relax	AM3b MRCA + Brazilian Wild A + B		All remaining branches	Selection Relaxation not significant (K = 0.94, *p* = 0.360, LR = 0.84)	
BUSTED	AM3b MRCA + Brazilian Wild A + B		All remaining branches	Evidence of gene-wide EDS (LRT, *p*-value = 0.022 ≤ 0.05)	

## Data Availability

Data is contained within the article or Appendix A. The data presented in this study are available in Appendix A.

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
