# Peer review of "A Novel Terrestrial Rabies Virus Lineage Occurring in South America: Origin, Diversification, and Evidence of Contact between Wild and Domestic Cycles"

_viruses, 2021, doi:10.3390/v13122484_

Round 1

Reviewer 1 Report

In this manuscript, by Caraballo et al., a novel rabies virus variant is described. The authors present a comprehensive analysis of phylogenetics and cross-species transmission in South America. The manuscript is well written, with detailed methodology and an in-depth and well-executed discussion of the results. As such, I only have a few minor comments

  1. Discussion, section 4.4, page 19, lines 231 onwards. The authors state that recombination is considered to be rare (line 231), however, the following sentence states that recombination is an important mechanism that drives RABV evolution. This seems to be the opposite of what is discussed in the next paragraph – only three putative recombination events were detected in the dataset of more than 200 sequences. Could the authors clarify how an event that is seen in approximately 1% of cases analyzed be considered an important driver for evolution?
  2. Line 235, the authors state that recombinants could lead to erroneous genetic typing. The authors should please define what is meant by “genetic typing.”
  3. Lines 236-238, the authors should please elaborate on what is meant by “unpredicted epidemiological results” and why this is important from a surveillance perspective

Author Response

Comment: Discussion, section 4.4, page 19, lines 231 onwards. The authors state that recombination is considered to be rare (line 231), however, the following sentence states that recombination is an important mechanism that drives RABV evolution. This seems to be the opposite of what is discussed in the next paragraph – only three putative recombination events were detected in the dataset of more than 200 sequences. Could the authors clarify how an event that is seen in approximately 1% of cases analyzed be considered an important driver for evolution?

Response: We understand the point raised by the Reviewer, and appreciate her/his efforts in evaluating our manuscript. It is true that recombination is not a ubiquitous mechanism in RABV evolution. What we intended to highlight is the fact that recombination is generally ignored in RABV surveillance and also in RABV evolutionary studies, and although infrequent, its absence has to be tested before conducting phylogenetic and/or selection analyses. We have rewritten this section, clarifying the inconsistency pointed by the Reviewer.

Comment: Line 235, the authors state that recombinants could lead to erroneous genetic typing. The authors should please define what is meant by “genetic typing.”

Response: We changed the expression to "Molecular typing", and added the following clarification: "Molecular typing in RABV surveillance is generally based on short sequences targeting the nucleoprotein gene. If recombination takes place within the sequenced fragment, it could lead to incorrect RABV variant identification".

Comment: Lines 236-238, the authors should please elaborate on what is meant by “unpredicted epidemiological results” and why this is important from a surveillance perspective

Response: An example of such unexpected results was included. The importance of detecting recombinants for surveillance was pointed out in the previous item.

Reviewer 2 Report

In this manuscript, the authors present a study looking into the evolution of Rabies viruses in South America particularly pertaining to the establishment of Antigenic variant 2 and adds to existing knowledge. The study is well conducted and sufficient detail regarding methodological approaches are provided as well as in data presentation. There are some minor issues which require clarification which I have outlined below.

Introduction

Lines 46-47

The authors clearly state the aim of the study here, but it would be useful to preempt this with a gap in knowledge in the understanding of evolutionary patterns and ecological interactions of the target RABV variant.

Materials and methods

Table 1

GenBank accession numbers are missing

GenBank accession numbers for sequences from the study submitted to GenBank are also missing. Have the sequences indeed been submitted to GenBank?

The software “Ray” should be clearly defined

The species Dusicyon was written as a stand alone the second time which is incorrect. “.spp” should be added.

The “3” in sampling generations 5x103 should be italicized for clarity.

The description of the phylogeographic analysis can be improved. Eg. the type of geographic units like country/region et used should be clearly stated

What was the purpose of conducting the network analysis? This should be made clear in the methods.

Author Response

Introduction

Lines 46-47

Comment: The authors clearly state the aim of the study here, but it would be useful to preempt this with a gap in knowledge in the understanding of evolutionary patterns and ecological interactions of the target RABV variant.

Response: We appreciate the Reviewer's effort in evaluating our manuscript. We have modified the corresponding paragraph to show the vacancy area on which the study was based.

Materials and methods

Table 1

Comment: GenBank accession numbers are missing

GenBank accession numbers for sequences from the study submitted to GenBank are also missing. Have the sequences indeed been submitted to GenBank?

Response: GenBank Accession numbers were obtained during manuscript revision. These are now included in Table 1 and the body of the manuscript.

Comment: The software “Ray” should be clearly defined

Response: a brief description of the algorithm implemented by the software Ray was added.

Comment: The species Dusicyon was written as a stand alone the second time which is incorrect. “.spp” should be added.

Response: It was corrected in all mentions, except for one in section 2.2, where we refer to the genus Dusicyon, and not to any particular species or form.

Comment: The “3” in sampling generations 5x103 should be italicized for clarity.

Response: We have applied superscript formatting to the power of ten.

Comment: The description of the phylogeographic analysis can be improved. Eg. the type of geographic units like country/region et used should be clearly stated

Response: This was stated in the Results section. In agreement with the Reviewer, we have moved it to the Materials and Methods section, and we delve into the rationale behind regionalization. In brief, regions represent separate units in terms of their biomes, weather, topography, hydrography, and vegetation.

Comment: What was the purpose of conducting the network analysis? This should be made clear in the methods.

Response: We have clarified the aim of conducting the network analysis. We used this approach to have a rapid representation of the spatio-temporal structure, as well as it helped us testing wether the geographic regionalization used was appropriate for Ancestral State Reconstruction. With this approach, we corroborated that samples were geographically structured within the regions used without exception.

Round 2

Reviewer 1 Report

Accept in present form

Reviewer 2 Report

The author's have adequately addressed all concerns previously raised and I have no further comments at this moment.